# Detailed characterization of neural selectivity in free viewing primates

Jacob L. Yates [1,2,3,4] ✉, Shanna H. Coop[1,2,5], Gabriel H. Sarch[1,6], Ruei-Jr Wu[2,7], Daniel A. Butts [3], Michele Rucci [1,2] & Jude F. Mitchell [1,2]

Fixation constraints in visual tasks are ubiquitous in visual and cognitive neuroscience. Despite its widespread use, fixation requires trained subjects, is limited by the accuracy of fixational eye movements, and ignores the role of eye movements in shaping visual input. To overcome these limitations, we developed a suite of hardware and software tools to study vision during natural behavior in untrained subjects. We measured visual receptive fields and tuning properties from multiple cortical areas of marmoset monkeys who freely viewed full-field noise stimuli. The resulting receptive fields and tuning curves from primary visual cortex (V1) and area MT match reported selectivity from the literature which was measured using conventional approaches. We then combined free viewing with high-resolution eye tracking to make the first detailed 2D spatiotemporal measurements of foveal receptive fields in V1. These findings demonstrate the power of free viewing to characterize neural responses in untrained animals while simultaneously studying the dynamics of natural behavior.

All animals with image-forming eyes acquire visual information through eye movements[1], which shapes the visual input[2]. However, standard characterizations of neural processing of vision, to date, require stabilization of the subject's gaze−either through anesthesia/paralytics[3,4] or trained fixation on a central point[5] (Fig. 1a)−or they ignore eye movements entirely[6]. Even experiments that involve active components of vision−such as covert attention, or the planning of saccadic eye movements−primarily involve analyses during instructed saccades and fixation on a point[7–9].

While these conventional paradigms have given us a highly successful model of early visual processing[10,11], it is unknown how well those results generalize to describe natural visual conditions and they have limited the study of visual processing to portions of the visual field outside the center of gaze. Relatively few labs have attempted to study visual processing during natural eye movements[12,13], and none have been able to interpret neural responses with respect to detailed visual processing in the presence of natural eye movements. Recent

experimental work has demonstrated that eye movements modulate neural selectivity substantially in many brain areas[14,15]. Moreover, visual input is normally acquired through eye movements. Recent work suggests this process is fundamental in formatting the visual input to facilitate normal vision[16–18]. Furthermore, fixation paradigms come with a substantial cost in our understanding of visual processing: the visual stimulus the subject is looking at (the fixation point) is not the stimulus under study[19,20].

To study natural vision without any loss of detail or rigor, we have developed a suite of integrated software and hardware tools to characterize neural selectivity during natural visual behavior, and do so at a resolution that exceeds standard fixation paradigms. Our approach, "free viewing", lets subjects look wherever they please within the visual display. We perform all analyses on a gaze-contingent reconstruction of the retinal input. Although previous studies have "corrected" for small changes in eye position by shifting the stimulus with the measured or inferred center of gaze, this has only been attempted for small

[1]Brain and Cognitive Sciences, University of Rochester, Rochester, NY, USA. [2]Center for Visual Science, University of Rochester, Rochester, NY, USA. [3]Department of Biology and Program in Neuroscience and Cognitive Science, University of Maryland, College Park, MD, USA. [4]Herbert Wertheim School of Optometry and Vision Science, UC Berkeley, Berkeley, CA, USA. [5]Neurobiology, Stanford University, Stanford, CA, USA. [6]Neuroscience Institute, Carnegie Mellon University, Pittsburgh, PA, USA. [7]Institute of Optics, University of Rochester, Rochester, NY, USA. ✉e-mail: yates@berkeley.edu

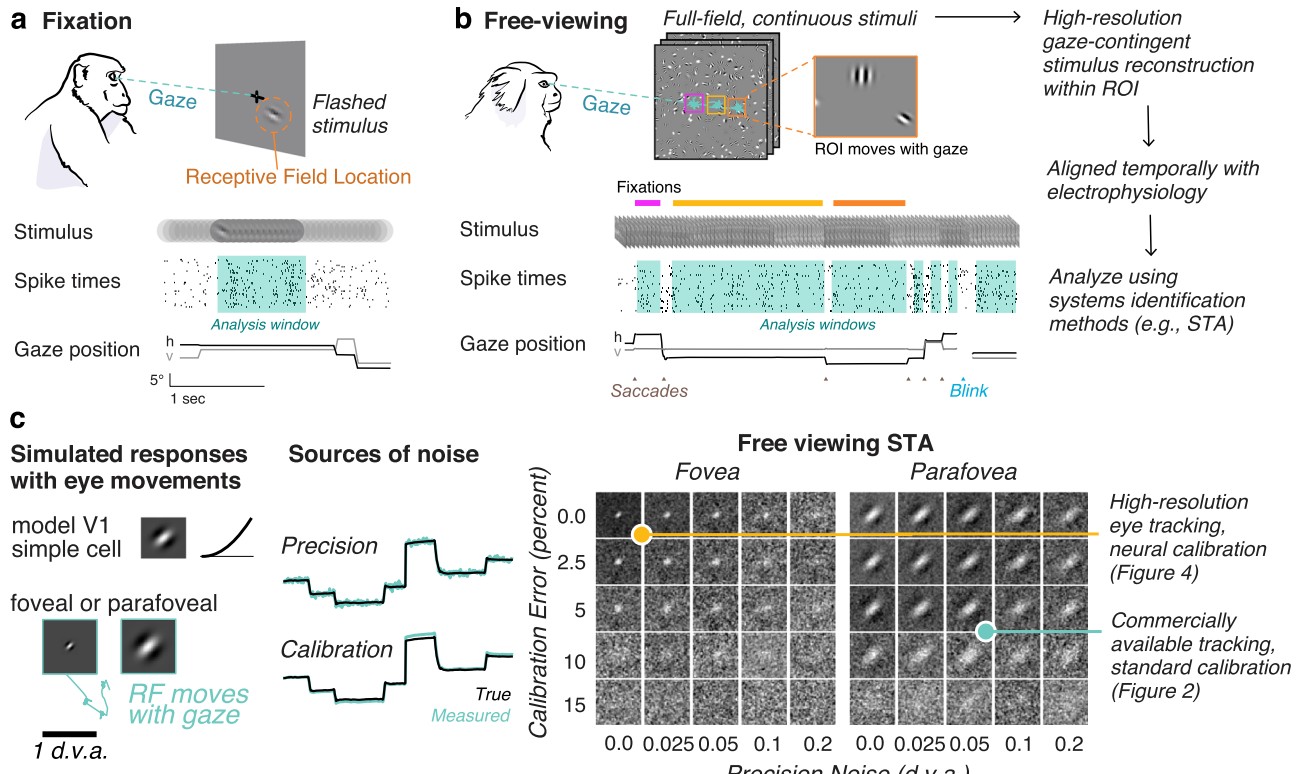

**Fig. 1 | Free-viewing paradigm and gaze-contingent neurophysiology.**
**a** Conventional fixation paradigm with flashed stimuli. Spike times are aligned to stimulus onset and analyzed during a window during fixation. **b** Free viewing: subjects freely view continuous full-field stimuli. Shown here is dynamic Gabor noise. All analyses are done offline on a gaze-contingent reconstruction of the stimulus with a region of interest (ROI). Analysis windows are extracted offline during the fixations the animal naturally produces. **c** Simulation demonstrates how uncertainty in the gaze position (due to accuracy and precision) would limit the ability to map a receptive field. (Left) A model parafoveal simple cell (linear receptive field, half-squaring nonlinearity, and Poisson noise) moves with the gaze. Example gaze trace shown in cyan. RF inset is 1 d.v.a. wide. (Middle) errors in

precision are introduced by adding Gaussian noise to the gaze position. Errors in calibration are introduced by a gain factor from the center of the screen. (Right) The recovered RF using spike-triggered averaging (STA) on a gaze-contingent ROI. Of course, with zero precision noise or calibration error, the STA recovers the true RF. Adding either source of noise degrades the ability to recover the RF, however, some features are still recoverable for a wide range of noise parameters at this scale. RFs that are smaller or tuned to higher spatial frequencies require high-resolution eye tracking. The yellow and cyan dots indicate two levels of accuracy that are explored in Figs. 2 and 4, respectively. Animal drawings in panel a and b were created with help from Amelia Wattenberger.

displacements of the stimulus during instructed fixation[21–23]. Our approach differs in that the subjects are free to explore the visual scene, resulting in more natural dynamics of visual input, and requiring little to no specific training.

Previous attempts at free viewing have faced three main obstacles: (1) the computational requirements to recover receptive fields from full-field stimuli, (2) limitations in eye tracker precision, and (3) errors resulting from eye-tracker calibration. Here, we show that combining image-computable neural models with correction for eye position from commercially available eye tracking is sufficient to recover receptive field size and tuning in free-viewing animals. Additionally, we introduce a high-resolution eye tracker for non-human primates and offline calibration using measured neurophysiology to give sufficient resolution to study visual processing in foveal neurons of primary visual cortex.

We demonstrate this approach here in marmoset monkeys. Marmosets are small new-world primates with homologous visual architecture to larger primates[24] and similar eye-movement statistics[25]. They are increasingly used as a model for neuroscience because of their similarity to humans and benefits for genetic tools[26]. However, marmosets are limited in their ability to fixate for prolonged periods. Our approach circumvents this issue, making both standard and new neural characterization approaches possible in marmoset, and also resulting in a higher data-throughput per animal: generating more data per unit time than fixation paradigms. More generally, this paradigm

provides an opportunity for rigorous study of visual neuroscience in species where fixation paradigms may be impractical (such as ferrets, tree shrews, and rodents).

We demonstrate the free-viewing approach to recover receptive field properties to both primary visual cortex (V1) where neurons can have high spatial and temporal resolution, and area MT, a higher visual area specialized for motion processing. Additionally, combined with high-resolution eye tracking, it is possible to recover fine-scale spatial receptive field structure of neurons in the central degree of vision (the foveola) for the first time.

## Results
### The free-viewing paradigm
To study natural vision in untrained animals, we depart from conventional approaches that stabilize the subject's gaze behaviorally with a fixation point. Instead, we present full-field natural and artificial stimuli in 20 s trials while monitoring eye position and neural activity. Figure 1b illustrates the free-viewing approach: because the retina moves with the eyes and visual neurons have receptive fields in a retinal coordinate frame, we must correct for changes in eye position to correctly represent the visual inputs to neurons. The relevant stimuli for a set of neurons can be recovered offline using a gaze-contingent region of interest (ROI) that moves with the eyes. Once the stimulus is reconstructed within the gaze-contingent ROI, conventional analysis tools can be used. In the following sections, we describe the successful

application of this approach to recordings from V1 and MT of 4 marmoset monkeys (*Callithrix Jacchus*; 3 males, 1 female).

A considerable barrier to using free-viewing paradigms prior to this work has been limitations in eye-tracking. Figure 1c demonstrates the effect of eye tracking limitations by simulating the responses of a model V1 simple cell with a receptive field that moves with the eyes and adding common sources of noise. Of course, if the true gaze position was perfectly known, the experimenter could recover the receptive field because the gaze-contingent input would be identical to the input with stabilized gaze. However, real eye trackers have noise that affects the precision of their measurements (top trace, precision). Eye trackers must also be calibrated, which is only as accurate as the subject's ability to fixate on points on the screen presented during the calibration procedure and has inherent error associated with it (bottom trace, calibration). Adding these sources of noise affects the ability of the experimenter to recover an RF from the free-viewing approach (Fig. 1c, right panel).

A second obstacle to free viewing is computational limitations in processing full-field high-resolution stimuli. A standard monitor today has $1920 \times 1080$ pixels. Generating artificial stimuli at high frame rates and full resolution is now possible with gaming graphics processing units (GPUs) and procedurally generated stimuli can be reconstructed offline at full resolution for part of the screen.

In the following sections, we show that tailored artificial stimuli combined with commercially available eye tracking can recover defining properties of neurons in V1 and MT. We then show that high-resolution eye tracking, combined with V1-based offline calibration can recover detailed spatiotemporal RFs in the fovea of V1.

## Retinotopy and selectivity in V1 during free-viewing paradigms

While the accuracy and precision of eye tracking impose limitations on the scale of receptive fields that can be studied during free viewing, a great deal can still be accomplished even with a standard eye tracker. In this section, we show that full-field sparse noise stimuli and commercially available eye tracking (Eyelink 1000) can be used to recover the size, location and tuning of receptive fields (RFs). The sparse noise stimulus allows us to efficiently estimate RF locations over a large portion of the visual field, which is often all that is required for further targeting neurons with behavioral paradigms, but also can be used to further target analyses with high-resolution stimuli within an ROI.

We present sparse noise consisting of flashed dots or squares in random positions on each frame (Fig. 2a) during free viewing and use a gaze-contingent analysis to align the stimulus to retinal coordinates. We move a grid with the location of gaze on each frame (Fig. 2a). As the grid moves with the eyes, we then average the stimulus luminance within each grid location on each video frame. This can be computed rapidly with sparse noise using the position and sign of the dots that are present on each frame of our noise stimulus. Our initial ROI is $28 \times 16$ degrees of visual angle (d.v.a.) with 1 d.v.a square bins, centered on the gaze location. This window covers a large portion of central vision, including all possible retinotopic locations in our recording chamber (and an equal area in the opposite hemifield).

We used regularized linear regression (methods) to estimate the spatiotemporal RF. As illustrated by the spatial response profile at the peak temporal lag (Fig. 2b), we can identify receptive fields at a coarse spatial scale (1 d.v.a. bins) for two example neurons: one in the fovea and one in the periphery. We then re-define a new ROI centered on simultaneously recorded RFs and run the same binning and regression procedure at a finer spatial scale with 20 bins (median = 0.2 d.v.a. bins) spanning the new ROI (Fig. 2b, insets). Spatial RFs were typically recovered with less than 5 min of recording time. The median recording time to recover spatial retinotopic maps was only 2.39 [2.03, 8.33] minutes ($n = 18$ sessions). We then fit a 2D gaussian to the fine-scale spatial map to recover the RF location and size (methods). Though it is possible to train marmosets to perform conventional

fixation tasks[25,27], much of that time would be unusable for analysis due to breaks between the trials and limited trial counts. Using the free-viewing approach here results in a substantial gain in the total analyzable neurophysiology data over fixation paradigms (Supplementary Fig. 1). We determined a unit had an RF if (a) the linear RF explained more variance on withheld data than the mean firing rate and (b) the Gaussian fit to the RF had an r-squared greater than 0.4. Using these criterion, the free-viewing analysis was sufficient to recover spatially selective RFs in 189/322 (58.7%) of recorded units with a sufficient number of spikes ($>200$) and physiologically reasonable spike waveforms (see methods) from marmoset V1 (Supplementary Table 1), and demonstrated a comparable relationship between eccentricity and size of RFs as reported from previous literature with anesthetized marmosets (Supplementary Fig. 2).

We also measured visual feature selectivity during free viewing using sinewave gratings. We presented full-field gratings that were updated randomly on each frame (Fig. 2c) and performed subspace reverse correlation[28] yielding the spatial-frequency RF for the same example units, plotted in polar coordinates where angle represents stimulus orientation and radial distance represents spatial frequency (Fig. 2d). We label units as having a significant RF following the goodness-of-fit criterion (a) and (b) from above, except the Gaussian fit was replaced with a parametric model of orientation and spatial-frequency tuning (see methods). This analysis produces selective responses in 377/428 (88%) of units (Supplementary Table 1) and worked well across the visual field. The subspace reverse correlation also gave temporal response functions consistent with known V1 temporal response profiles (Fig. 2e). The median recording time used for grating receptive fields was 11.03 [10.66, 16.33] minutes ($n = 18$ sessions). The resulting distribution of preferred orientations (Supplementary Fig. 2) was comparable to previous reports from macaque and cat V1[29,30]. Thus, the feature tuning of neurons in V1 can be measured during free viewing with short recording times in minimally trained marmosets using commercially available eye tracking and standard calibration.

## Free-viewing approach recovers receptive field properties in area MT

The validity of this approach is not limited to simple visual features that drive primary visual cortex, but can generalize to other features and higher level visual areas. We demonstrate this here by measuring motion-selective RFs for neurons recorded from area MT during 10 free-viewing sessions. Extra-striate area MT is a higher-order visual area with the vast majority of neurons exhibiting exquisite tuning to retinal motion[31]. To measure motion-selective RFs, we adapted the sparse noise stimulus described above to include motion. Rather than simply appearing and disappearing on each frame, each dot was randomly placed with asynchronous updating and then drifted drifted for 50 ms in one of 16 directions (Fig. 3a). Using the same gaze-contingent analysis window, we converted the spatiotemporal stimulus into separate horizontal and vertical velocity components. This spatiotemporal velocity stimulus was then used as the input to a linear nonlinear poisson (LNP) model of the MT neuron spike trains (methods).

The LNP model trained on gaze-contingent velocity stimuli recovered spatiotemporal velocity RFs for MT units (Fig. 3b). We found detailed spatiotemporal measurements of the velocity selectivity of MT neurons, which we decomposed into spatial maps of direction selectivity (Fig. 3b), temporal selectivity (Fig. 3c), and overall motion tuning (Fig. 3d, see methods). Following the selection criterion above, we labeled neurons as having selective RFs if the linear RF cross-validated better than the mean firing rate. This yielded 241/466 (51.72%) selective units (Supplementary Table 2). The selective MT neurons in our sample were well fit by von Mises tuning curves (mean r-squared = 0.62 + −0.02, $n = 241$ units). This highlights that full-field stimuli can be engineered to target complex feature

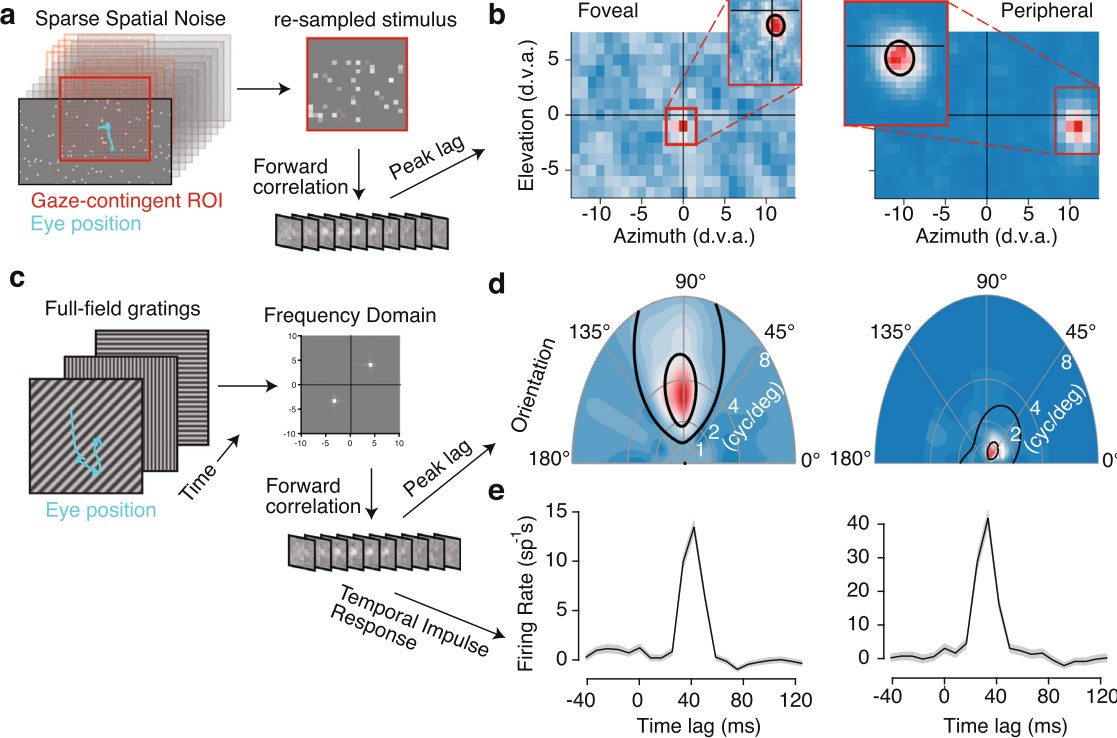

**Fig. 2 | Receptive field mapping and feature tuning in V1. a** Retinotopic mapping approach uses a 28 × 16° gaze-contingent grid with 1° spacing. This downsampled stimulus is used to estimate the receptive field (RF) using linear regression with binned spike counts. **b** The spatiotemporal RF at the peak lag is shown as a heatmap for foveal and peripheral example units. A region of interest (ROI) is set around the peak value in the RF map. This ROI is used to repeat the analysis at a .3° resolution, which is shown in the insets. The black line denotes a contour from Gaussian fit at 2 standard deviations. **c** Subspace reverse correlation procedure for mapping

tuning. Full-field gratings are updated randomly at or near the frame rate. Linear regression is used to map from the frequency subspace to firing rates. The resulting spatiotemporal weights are used to learn the tuning and temporal impulse response. **d** Example joint orientation-frequency tuning maps for the units in panel **b**. Black lines indicate 50 and 75% contour lines from the parametric fit (methods). **e** Temporal impulse response in spikes per second for preferred gratings measured with forward correlation.

selectivity and that regression-based analyses can recover detailed spatiotemporal measurements of that selectivity during uncon-strained visual behavior.

### High-resolution eye tracking for detailed 2D spatiotemporal receptive fields in the fovea

No previous studies have accurately recovered the full spatiotemporal receptive field structure of V1 neurons in primate foveal regions. This gap is not due to negligence, but rather reflects limitations in the accuracy of eye tracking in measuring small RFs. Even anesthetized, paralyzed monkeys exhibit drift in eye position over the course of an experiment. Further, the conventional approach to obtain accurate RF estimates using fixation paradigms obscures study of foveal vision because the center of gaze is occupied by the fixation point as the stimulus. Another limitation to all previous studies is that fixation is imperfect, with continuous eye drift and fixational eye movements, which are substantial and would limit precision if uncorrected (Sup-plementary Fig. 3). The free-viewing approach provides an opportunity to directly stimulate foveal vision to recover high-resolution RFs if it used in conjunction with sufficiently accurate eye-tracking. Here, we apply free viewing with high-resolution (both in terms of accuracy and precision) eye tracking to measure detailed receptive fields in the fovea of free-viewing marmosets.

To obtain precise measurements of gaze position, we adapted a recently developed video eye tracker[32,33] for use with marmosets. The digital Dual Purkinje Imaging (dDPI) eye tracker uses a digital CCD camera, IR illumination and GPU processing to track the 1st and 4th Purkinje images achieving a 0.005 degree precision (RMS of noise measured with an artificial eye) and is precise enough to

measure and correct for fixational drift and microsaccades (Supplementary Fig. 3).

To achieve full-resolution receptive fields in the fovea, we center an ROI on the retinotopic location of the recorded neurons (as in Fig. 2) and then reconstruct the full stimulus (pixel by pixel) for every frame within that ROI (as in Fig. 1b). Beyond flashed spatial dots or gratings, we also presented trials in which the free-viewing background consisting of flashed Gabor and Gaussian stimuli of varying phase, orientation, and spatial scales. For 5 foveal recording sessions, we reconstructed every frame of the experiment at pixel-resolution (where pixels were 1.5 arcmin) within a gaze-contingent ROI. This was done for every stimulus condition so that we had a gaze-contingent movie of the stimulus at the projector refresh rate (240 Hz).

While the dDPI tracker used in the current study provided high precision position signals, it required calibration to obtain an accurate estimate of the actual gaze position. To calibrate high-resolution eye-trackers, previous studies in humans use a two-stage calibration pro-cedure, where the human subjects adjust their own calibration para-meters in a closed loop[34]. As our marmosets were unlikely to perform self-calibration without extensive training, we developed an offline calibration method using V1 physiology directly. Briefly, we fit a con-volutional neural network (CNN) model of V1 that included a recali-bration of the eye tracker to optimize the model fits to gaze-contingent neural activity across the recorded session. Specifically, the CNN pre-dicts the spiking response of the entire population of simultaneously recorded units given the spatiotemporal gaze-contingent stimulus movie, which is reconstructed based on the eye tracker calibration identified by the CNN (Fig. 4a). One critical aspect of this method is

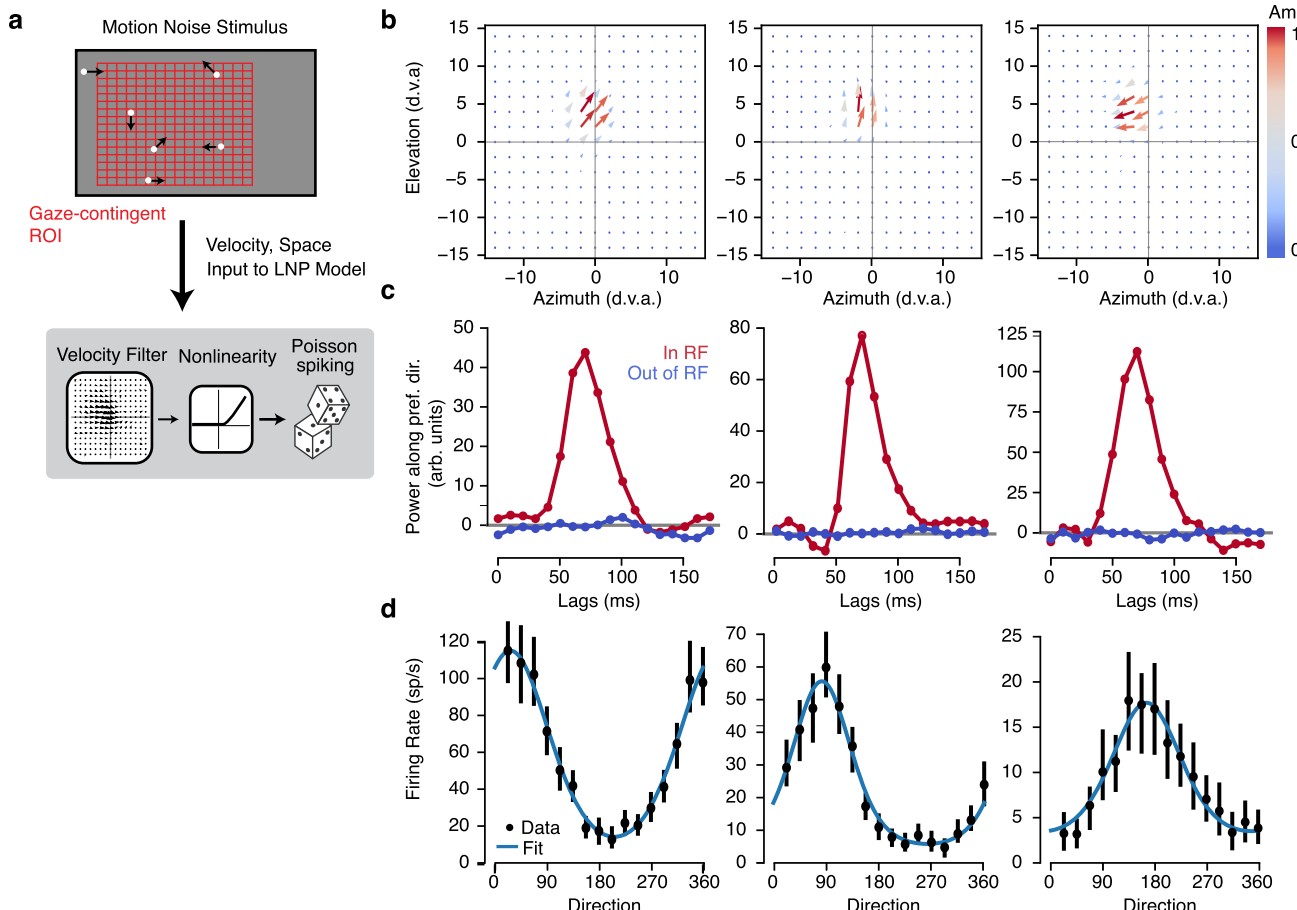

**Fig. 3 | Receptive field mapping and tuning in MT. a** Sparse motion-noise stimulus was converted into a spatiotemporal velocity stimulus with separate horizontal and vertical velocities using a gaze-contingent grid with 2° spacing. This downsampled stimulus is used to estimate the receptive field (RF) using linear nonlinear Poisson model (LNP). **b** The spatial map at the peak lag of the spatiotemporal velocity RF from the LNP fits is shown as a vector plot for three example units. Color indicates the vector amplitude (ranging from 0 to 1, with gray at 0.5). **c** The temporal impulse response was measured both in and out of the RF by projecting the (unnormalized) vector at the maximum and minimum amplitude of the spatial RF on the preferred direction (unit vector). The three plots correspond to the three example units in **b**. **d** Tuning curves were measured by masking the stimulus with half of the max of the spatial RF and computing the mean firing rate at the peak lag for each direction shown. Error bars are 95% confidence intervals measured with bootstrapping and blue lines are fits with a von mises function.

that we obtain better fits to the eye tracker calibration when we use recordings from larger V1 populations. Therefore, using recordings with high-density arrays as used in this current study represents a distinct advantage. Importantly, the CNN is only used to learn the correction grid for the eye-tracker calibration. The RFs we measure here are conventional spike-triggered averages (STAs) and are not a product of the CNN.

The key component of the CNN that corrects the calibration of our eye tracker is a 2-layer "shifter" network[35]. The shifter network takes the gaze position on each frame as input and produces a *shared* shift to the spatial position of *all* units being reconstructed in our recording session (Fig. 4a). Any errors in the behavioral calibration of the eye tracker will manifest as shifts in the RF locations for all units as a function of where the marmoset is looking, which will be learned by the shifter network. Such corrections to the output of the dDPI eye tracker end up being smooth functions of spatial position, as illustrated by the output of the shifter network for an example session visualized as a function of gaze position (Fig. 4b). As was typical of our sessions, the shifter network produced small shifts (maximum 7 arcmin, median shift 1 arcmin) across a 10 d.v.a. range of gaze positions. These calibration matrices (CMs) dictate how to correct the initial experimental calibration. Across repeated fitting, the shifter network produced highly reliable CMs, suggesting that the shifter was measuring systematic changes in RF position as a function of the monkey's

gaze position, likely both a result of small errors in the initial calibration established by the experimenter, and systematic deviations from linearity of the calibration.

Once an accurate calibration was established, we reconstructed the detailed retinal input to the receptive field at foveal precision. To measure the RF, we computed the spike-triggered average (STA) stimulus. To avoid potential issues of post-saccadic lens wobble[36], we excluded 50 ms following each saccade from our analyses. We compared the linear RF computed with spike-triggered averaging (STA) with and without the improved CM correction. Figure 4c shows example foveal RFs measured with and without correction. The examples in Fig. 4c illustrate that the shifter network is necessary to make detailed measurements in the fovea, as reflected by the RFs measured with and without correction. RFs post correction had amplitudes that were significantly larger than without correction (geometric mean ratio = 1.38 [1.34, 1.41] $n = 378$, $p < 2.12 \times 10^{-209}$, 1-sample $t$-test $t = 66.08$).

To calculate whether a unit had a significant RF, we compared our measured STAs to a null distribution computed using spikes from before the stimulus onsets. Using a $p < 0.001$ threshold, 378/621 (60.87%) of units had significant RFs. We found that at this scale, many more units had significant RFs using squared pixel values (572/621 [92.11%]) suggesting they are visually responsive, but not well described by a linear RF. Because errors in measured gaze position may

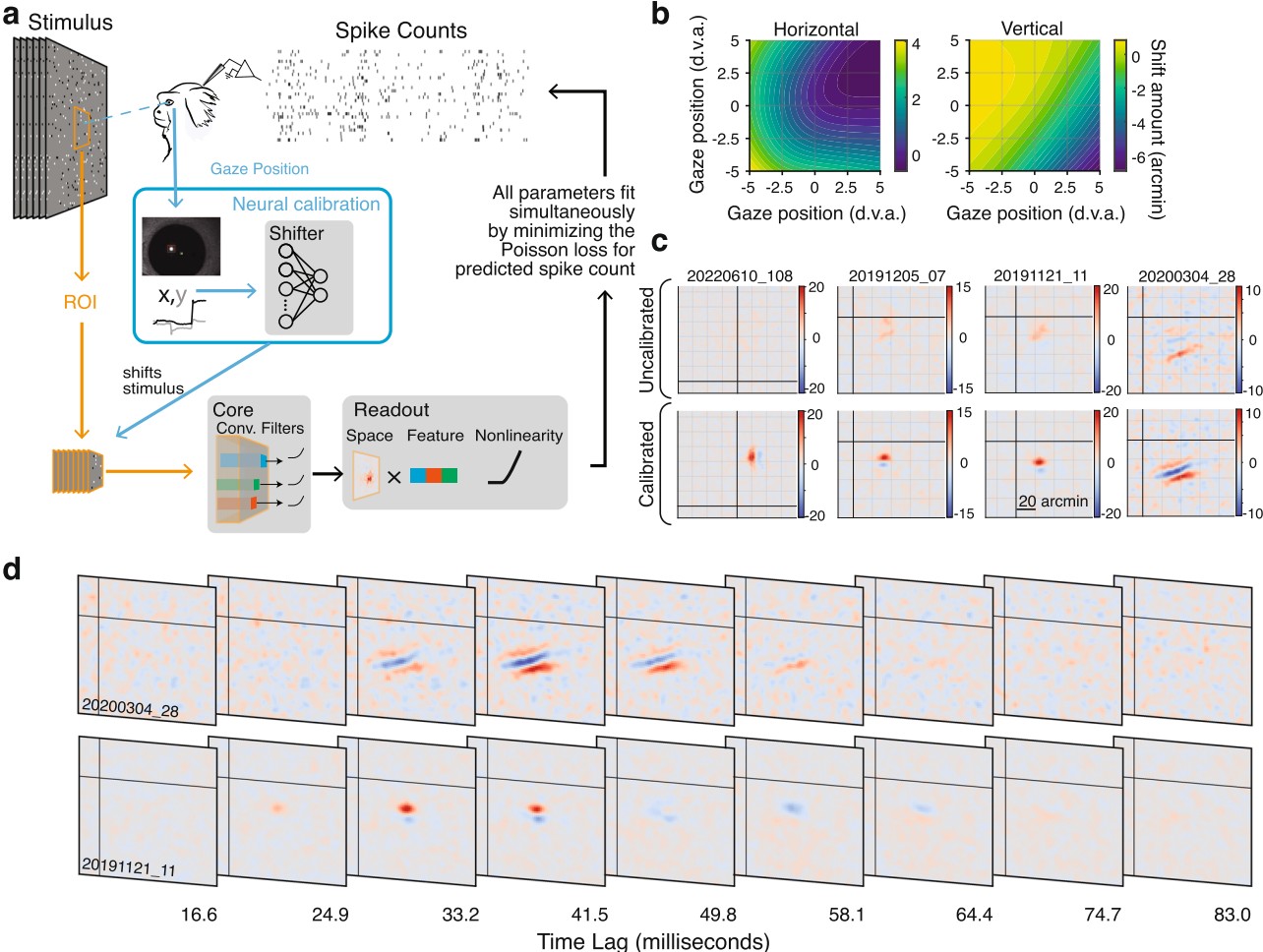

**Fig. 4 | Neural eye-tracker calibration and high-resolution foveal receptive fields. a** Convolutional Neural Network (CNN) architecture used to calibrate the eye tracker. The gaze-contingent stimulus within the ROI is processed by the nonlinear subunits of the convolutional "Core". The "Spatial Readout" maps from the core to the spike rate of each neuron with a spatial position in the convolution and a weighted combination of the feature outputs at that position. This is passed through a static nonlinearity to predict the firing rate. The "Shifter" network takes in the gaze position on each frame and outputs a shared shift to all spatial readout positions during training. All parameters are fit simultaneously by minimizing the Poisson loss. After training, the shifter output is used to shift the stimulus itself so further analyses can be done on the corrected stimulus. **b** Calibration correction grids for horizontal and vertical gaze position are created using the output of the shifter network for one session. These are used to correct the stimulus for further analysis. **c** Spatial receptive fields measured with spike-triggered average before (top) and after (bottom) calibration for four example foveal units demonstrates the importance of calibration for measuring foveal RFs. RFs were z-scored and plotted on the same color scale before and after calibration. Grid lines are spaced every 20 arcminutes. **d** Example foveal spatiotemporal RFs. The axes bounds are the same as in **c**. In both **c** and **d**, black lines indicate the center of gaze. Marmoset drawing in panel **a** was created with help from Amelia Wattenberger.

manifest as a scrambling of the phase of the input, which would result in a bias towards finding complex cells, we investigated the proportion of simple and complex cells in our sample by fitting simple and complex cell models to each cell and comparing the cross-validated log-likelihoods (Supplementary Fig. 4). Using this method, we found 81% of our units were complex, consistent with previous reports from anesthetized marmoset measured outside the fovea[37].

Our recordings from marmoset V1 are the first detailed 2D spatiotemporal measurements of foveal cortical processing. Figure 4d shows example single neuron spatiotemporal RFs. Across neurons, we found a range of spatial and temporal receptive field properties that are consistent with several classic findings of simple and complex cell receptive field structure in V1, provided a substantial miniaturization for the spatial scale.

To compare the RFs recovered during free viewing with fixation more directly, we recorded the same units during fixation and free viewing. Foveal RFs were too small to be recovered during standard fixation without correcting for measured eye movements. However, if we corrected for miniature eye movements made

during fixation, we could still recover foveal RFs, and in that case we find a close correspondence to the RF recovered during free viewing (Supplementary Fig. 5).

These preliminary findings decisively demonstrate the power of the free-viewing methodology when combined with high-resolution eye tracking and neural eye tracker calibration. They also open a new avenue of research for examining foveal scale visual representations not only in V1 but also other visual area along the ventral processing stream which specialize in higher acuity object vision.

## Discussion
We introduced a free-viewing paradigm for visual neuroscience. This approach is higher-yield (per unit recording time) than fixation-based approaches for marmosets and yields measurements of spatial RFs and feature tuning in minimally trained animals. It works with standard commercially available eye trackers for standard descriptions in V1 (Fig. 2) and MT (Fig. 3). We also demonstrated this paradigm can be extended to study foveal V1 neurons by introducing a high-resolution video eye tracker based on the dual Purkinje method and a calibration

routine based on the output of V1 neurons. Combined with free viewing, these methods are state-of-the-art in the measurement of 2D spatiotemporal receptive fields of neurons in the foveal representation of V1 (Fig. 4). Thus, the three immediate advantages to free viewing, especially when combined with high-resolution eye tracking, are: (1) application to untrained animals, (2) increased usable data per unit recording time, and (3) measurement of foveal visual processing. Further, although we only analyzed epochs of stable fixation between saccades in the present study, the free-viewing paradigm will also have major advantages for studying the role of eye movements during visual processing.

Free viewing is amenable to any animal model with almost no training requirements. Here, we applied this approach to marmosets. In the last decade, mice have emerged as a popular model for visual neuroscience[38]. Despite the fact that mice move their eyes in a directed manner[39,40], the spatial position of gaze is rarely accounted for in neurophysiological studies of visual cortex in mouse. As notable exceptions to this, Lurz et al. and Parker et al. used a similar shifter model to the one we employ to model mouse V1[35,41]. The free-viewing paradigm is a potentially promising future direction to expand rigorous visual neuroscience to animal models with higher acuity and smaller RFs than mice, but without the ability to perform trained fixation (such as ferrets and tree shrews). This type of paradigm will also support a direct comparison of visual processing and modulatory signals in multiple species, such as the role of locomotion in visual processing for non-human primates.

Offline gaze-contingent analysis of neural data during free viewing opens the possibility of studying neural computations in a range of natural visual behaviors and exceeds the resolution set by fixation studies. Although, previous studies have corrected for small changes in gaze position by shifting the stimulus with the measured or inferred center of gaze, this has only been attempted for small displacements of the stimulus during instructed fixation[21–23]. Our approach differs in that the subjects are free to explore the visual scene, and therefore, both the calibration and the displacements must be accounted for. Importantly, our use of visual cortex to calibrate the eye tracking differs from previous approaches such as neural-based eye tracking[22] in that all of the temporal dynamics of gaze are directly measured by a physical eye tracker that is independent of receptive field properties, as opposed to being dynamically inferred from neural activity. The CNN used neural activity only to improve the calibration of the eye tracker, and played no role in subsequent analyses of receptive fields, although its abilities to more robustly predict nonlinear processing in V1 will be an advantage to future studies. The added precision also makes it possible to examine the role of fixational drift and eye movements, an essential component of vision[16]. Further studies could assess, for example, whether RFs in V1 are explicitly retinotopic or dynamically shift to account for small fixational movements as proposed by recent theoretical work[42]. And as illustrated in Fig. 4, this approach affords the opportunity to examine foveal receptive fields in primate V1 for the first time. Despite its paramount importance for human vision, almost nothing is known about neural processing in the foveal representation.

Throughout the manuscript, we have highlighted that free-viewing paradigms can be used to recover receptive field properties in multiple areas. Although the resulting receptive fields and tuning properties are consistent with previous findings in V1 and MT, we did not systematically compare fixation and free viewing except in one session (Supplementary Fig. 5). Interestingly, we were not able to recover foveal RFs using conventional fixation approaches and, even in the fixation condition, we had to correct for positional shifts due to fixational eye movements. These results suggest that substantial variability in the central visual field is driven by fixational eye movements and future work should examine exactly how ignoring that

variability has contributed to estimates of signal and noise correlations[23] and how that variability contributes to the encoding process in natural visual conditions.

Finally, one limitation of our approach is by letting the eye's move freely, there are no longer repeats of the same stimulus condition, which is one of the main workhorses of systems neuroscience. However, fixational eye movements preclude that reality, even where it has been used previously[23]. With the development of higher speed cameras and video displays, it will soon be possible to stabilize retinal images during free viewing, thus affording more precise control of the stimuli input to visual neurons than previously possible in fixation paradigms. Examining natural behaviors that lack fixed repetitions is possible with comparable rigor as conventional approaches as shown here, when using appropriate neural models to fit the responses to natural stimuli. In near future, the application of neural models during natural behavior will finally allow us to gain deeper insight into the dynamics of neural processing in natural contexts.

## Methods

### Surgical procedures

Data were collected from 4 adult common marmosets (*Callithrix jacchus*; one female and three males). All surgical and experimental procedures were approved by the Institutional Animal Care and Use Committee at the University of Rochester in accordance with the US National Institutes of Health guidelines. At least one month prior to electrophysiological recordings, marmosets underwent an initial surgery to implant a titanium headpost to stabilize their head during behavioral sessions[27].

A second surgery was performed under aseptic conditions to implant a recording chamber. For the chamber implantation, marmosets were anesthetized with intramuscular injection of Ketamine (5–15 mg/kg) and Dexmedetomidine (0.02-0.1 mg/kg). A 3D-printed chamber (http://www.protolabs.com) was then attached to the skull with metabond (http://www.parkell.com) over coordinates guided by cranial landmarks. A 3×4 mm craniotomy was then drilled within the chamber (http://www.osadausa.com). The dura was slit and exposed tissue was covered with a thin layer (<2 mm) of a silicone elastomer (World precision instrument, https://www.piinc.com) as in Spitler and Gothard (2008)[43].

### Electrophysiological recordings

Electrophysiological recordings were performed using multisite silicon electrode arrays. The arrays consisted of 1–2 shanks, each containing 32 channels separated by 35 or 50 μm. The electrode arrays were purchased from NeuroNexus (http://www.neuronexus.com) and Atlas Neuro Engineering (https://www.atlasneuro.com). We recorded from neurons using a semi-chronic Microdrive system. We adapted the EDDS Microdrive System (https://microprobes.com) for use with silicone arrays and to be removable. Our chamber and drive designs are available online (https://marmolab.bcs.rochester.edu/resources.html). A reference wire was implanted under the skull at the edge of the chamber. The electrode arrays were lowered through the silicone elastomer and into brain using a thumbscrew.

Data were amplified and digitized at 30 kHz with Intan headstages (Intan) using the open-ephys GUI (https://github.com/open-ephys/plugin-GUI). The wideband signal was highpass filtered by the headstage at 0.1 Hz. We corrected for the phase shifts from this filtering (Okun, 2017). The resulting traces were preprocessed by common-average referencing and highpass filtered at 300 Hz. The resulting traces were spike sorted using Kilosort or Kilosort2. Outputs from the spike sorting algorithms were manually labeled using 'phy' GUI (https://github.com/kwikteam/phy). Units with tiny or physiologically implausible waveforms were excluded.

## Eye-tracking and saccade detection

Gaze position was monitored using one of two eye-trackers. 28 sessions did not involve high-resolution measurements in the foveal representations. Gaze position was sampled at 1000 Hz using an Eyelink 1000 (SR Research). For 30 high-resolution sessions, a custom digital Dual-Purkinje Imaging system (dDPI) was used. The dDPI uses a collimated IR beam (ThorLabs) a dichroic mirror (Edmunds) and samples 0.4 Megapixel images of the eye at 539 frames per second (DMK 33UX287; The Imaging Source). Custom CUDA code running on a gaming GPU (GTX 1080Ti; Nvidia) performs the algorithm described in refs. 32, 33 to extract the gaze position. Briefly, the 1st and 4th Purkinje images (P1 and P4) were identified and tracked using a two-stage process. The first stage is to find the region of interest (ROI) for each. The camera image is downsampled by a factor of 4 and P1 is found by thresholding the 8-bit image at 200 and calculating the center of mass of the pixels exceeding the threshold. The ROI for P4 was found via template matching on the downsampled frame. Following the initial ROI finding stage, the center of each Purkinje image was calculated using the full-resolution image within each ROI by center of mass for P1 and radial symmetric center for P4 (cite).

Methods for calibrating both eye-trackers before a behavioral session were identical to those described previously[25,27]. Briefly, this procedure sets the offset and gain (horizontal and vertical) of the eye-tracker output manually. The calibration was refined offline using a bilinear regression between the gaze position during a detected fixation and the nearest grid target (within a 1 degree radius) during the calibration routine.

Saccadic eye movements were identified automatically using a combination of velocity and acceleration thresholds as described in ref. 44. The raw gaze position signals were re-sampled at 1 kHz, and horizontal and vertical eye velocity signals were calculated using a differentiating filter. Horizontal and vertical eye acceleration signals were calculated by differentiation of the velocity signals using the same differentiating filter. Negative going zero crossings in the eye acceleration signal were identified and marked as candidate saccades. These points correspond to local maxima in the eye velocity signal. Eye velocity and acceleration signals were then examined within a 150 ms window around each candidate saccade. Candidate saccades were retained provided that eye velocity exceeded 8°/s and eye acceleration exceeded 2000°/$s^2$ . Saccade start and end points were determined as the point preceding and following the peak in the eye velocity signal at which eye velocity crossed the 10°/s threshold.

## Visual stimuli and behavioral training

For all V1 recording sessions, visual stimuli were presented on a Propixx Projector (Vpixx) with a linear gamma. The luminance of the projector ranged from 0.49 to 855 $cd/m^2$ with a mean gray of 416 $cd/m^2$. All stimuli were generated in Matlab (the Mathworks) using the Psychtoolbox 3[45]. Stimulus and physiology clocks were aligned and synchronized using a Datapixx (Vpixx) to strobe unique 8-bit words to the Open Ephys system. Stimulus code is available online at (https://github.com/jcbyts/MarmoV5).

## Foraging task

All visual protocols besides the static natural images were run simultaneously with a "foraging" paradigm where marmosets obtained a small juice reward (marshmallow water) for fixating small (0.5–1.0 d.v.a diameter) targets that would appear randomly in the scene. Reward was granted any time the marmosets kept their gaze within a specified radius of the center position of the target for more than 100 ms. Targets consisted of either oriented Gabor patches (2 cycle/deg) or marmoset faces that were taken from photos of the colony. A single face or Gabor target was presented at all times. Once the target was acquired for liquid reward it was immediately replotted at a new location

allowing the task to continue. Marmosets will naturally look at faces[25] and these were used to encourage participation in the forage paradigm. The position of the targets was generated randomly near the center of the screen (either drawn from a 2D Gaussian at the center or from an annulus with a 3 d.v.a. radius) to encourage the gaze to stay near the center of the screen where eye-tracking accuracy and precision are highest. The amount of reward was titrated based on the subject's performance to ensure they did not get too much marshmallow in a single session (5–10 μl per reward).

## V1 retinotopic mapping and receptive field size

Retinotopic mapping stimuli consisted of full-field randomly flashed full-contrast circles or squares (referred to as "dots" from here on). Each dot was either white (855 $cd/m^2$) or black (0.49 $cd/m^2$), and appeared at a random position anywhere on the screen. Across sessions, the dot-size and number of dots per frame varied, but were fixed within a session.

Offline gaze-contingent retinotopic mapping was performed in a two-stage process using regularized linear regression[46]. First, we estimated the RF at a coarse resolution and then re-sampled the stimulus at a finer resolution within an ROI centered on the result of the first stage. The coarse resolution RF was created by re-sampling the dot stimulus on a gaze-contingent grid. This rectangular grid, $G_{x,y}$, consisted of 405 locations spaced by 1 d.v.a. from −14 to 14 d.v.a along the azimuthal axis and −8 to 8 d.v.a. of elevation. The re-sampled gaze-contingent stimulus is a vector $X(t)$ at frame $t$ and was calculated by summing over the dots on each frame

$$X_{x,y}(t) = \sum_i^N f(D_i(t) - G_{x,y}) \tag{1}$$

where $D_i(t)$ is a vector of the position of the i-th dot on frame $t$, and $f$ is a function that returns a vector of zeros with a 1 at the grid location where the dot was centered. This method is fast and does not require regenerating the full stimulus at the pixel-resolution. Additionally, by summing the number of dots within each grid location, this analysis ignored the sign of the dot ("black" or "white") relative to the gray background, which was designed to target cells that exhibited some phase invariance (i.e., complex cells). We found that ignoring sign generated more robust retinotopic mapping results with fewer datapoints. We included neural recordings for analyses from mapping stimuli that ranged widely in their duration, provided at least one minute of data was collected.

We estimated the spatiotemporal receptive, $Ksp$, of each unit, $i$, by using regularized linear regression between the time-embedded gaze-contingent stimulus $X$ and the mean-subtracted firing rate, $R$, of the units binned at the frame resolution.

$$Ksp_i = \left(X^T X + \lambda D\right)^{-1}\left(X^T R_i\right) \tag{2}$$

Where $D$ is a graph Laplacian matrix corresponding to spatial and temporal points in $X$ and $\lambda$ is a scalar that specifies the amount of regularization. $\lambda$ was chosen using cross-validation. This measures the spatiotemporal receptive field (RF) in units of spikes per second per dot. We then repeated this processes at smaller grid size centered on the RF location recovered from the coarse stage. We found the RF location by thresholding $Ksp$ at 50% of its max an used the matlab function regionprops to find the centroid and bounding box. We scaled the bounding box by 2 and set 20 bins to span that region, with a bin size between 0.1–0.3 d.v.a. (median = 0.2 d.v.a.). We then re-ran the regression analysis to estimate the final RF. We then fit a 2D Gaussian to the spatial slice at the peak lag using least-squares with a global search over parameters and multiple starts. We converted the fitted covariance matrix to RF area using $area = \pi s_1 s_2$, where $s_1$ and $s_2$ are

the sqrt of the eigenvalues of the covariance matrix. This reports the area of an ellipse at 1 standard deviation.

Units were excluded if the mean shifted by more than 0.25, meaning that the fitting procedure produced a Gaussian that was not centered on the RF centroid. This resulted in 410/739 units with measurable spatial RFs.

## V1 tuning
To measure the neurons' selectivity to orientation and spatial frequency, we flashed full-field sinewave gratings. Gratings were presented at 25% contrast and were either drawn from the Hartley basis[47] or were parameterized using a polar grid of orientations and spatial frequencies. On each frame, only one grating was presented and up to 50% of the frames were a blank gray background. We represented the frequency space on a polar basis. The basis consisted of 8 evenly spaced von mises functions for orientation, and 4 nonlinearly stretched raised cosine functions for spatial frequency. This basis served to convert stimuli collected with the Hartley set and the polar grid to the same space. We measured the grating receptive field $Kgrating$ for each unit using regularized linear regression (as described above for the spatial mapping).

To measure the tuning curve of the units, we fit a parametric model of the form:

$$Kgrating(\theta, \omega) = b + (M - b)e^{-\kappa \cos^2(\theta - \hat{\theta}) - 1}e^{\frac{-(\log(1+\omega) - \log(1+\hat{\omega}))^2}{2\sigma^2}}/(e^\kappa - 1) \quad (3)$$

Where $\theta$ is the orientation, $\omega$ is spatial frequency, $\hat{\theta}$ and $\hat{\omega}$ are the orientation and spatial-frequency preference, respectively; $b$ is the baseline firing rate, $M$ is the maximum firing rate, $\kappa$ and $\sigma$ scale the width of the orientation and spatial-frequency tuning. This parametric form combines a normalized von Mises tuning curve for orientation that wraps every $\pi$ and a log-gaussian curve for spatial-frequency tuning (each normalized to have a maximum of 1 and a minimum possible value of 0). We converted the dispersion parameters into bandwidths as the full-width at half height of each curve. Tuning curves were fit using nonlinear least-squares (lsqcurvefit in matlab).

## MT velocity receptive fields and direction tuning
MT mapping stimuli consisted of sparse dot motion noise. Every video frame contained up to 32 white dots that were 0.5 degrees in diameter. Each dot was either replotted randomly or moved at 15 degrees/s in one of 16 uniformly spaced directions with a lifetime of 5 frames (50 ms with frame rate at 100 Hz). Marmosets performed the foraging task while this motion-noise stimulus ran in the background.

To calculate the RFs, the dot displacement on each frame transition was split into horizontal and vertical velocity components at each spatial location on a gaze-contingent grid with 2 d.v.a. wide bin size. This produced two gaze-contingent spatiotemporal stimulus sequences of the same form as described for V1 retinotopic mapping methods separate for horizontal and vertical velocities. Velocity receptive fields were measured by fitting a Poisson Generalized Linear Model (GLM) to the spike trains of individual MT units. The parameters of the GLM include the RF of the unit and a bias parameter to capture baseline firing rate. The RF parameters were penalized with to support spatial smoothness and sparseness using the same Graph Laplacian penalty used for retinotopic mapping and an L1 penalty. Example fitting and analysis code is available at https://github.com/VisNeuroLab/yates-beyond-fixation.

To measure the temporal integration of MT RFs, we first computed the "preferred direction vector" of the unit as the weighted average of the recovered RF at the peak lag. We then found the spatial location with the largest amplitude vector and calculated the projection of the RF direction vector at that location onto the preferred

direction vector for all time lags. We repeated this for the spatial location with the smallest amplitude vector.

To measure the direction tuning curves, we masked the stimulus spatially at every location greater than half of the max of the spatial RF and counted the number of dots drifting in each direction on each frame. We then calculated the direction-triggered firing rate of each unit through forward correlation between the directions on each frame and the firing rate, normalized by the number of dots shown. The tuning curve was taken to be the value for each direction at the peak lag. Error bars were computed using bootstrapping and correspond to 95% confidence intervals. We fit a von Mises function to the firing rate $R$

$$R = b + A \exp\left(K\left(\cos\left(\theta - \hat{\theta}\right) - 1\right)\right) \quad (4)$$

Where $b$ is the baseline firing rate, $A$ is the amplitude, $K$ is the bandwidth and $\hat{\theta}$ is the preferred direction.

## Full-resolution stimulus reconstruction
Stimuli were reconstructed by playing back the full experiment. All randomized stimuli were reconstructed using stored random seeds and the replayed frames were cropped within the gaze-contingent ROI using using Psychtoolbox function Screen('GetImage'). Randomized stimuli included flashed dots described earlier, as well as flashed Gabor noises that will be described in detail below. All high-resolution analyses operated on this reconstruction. For the 5 sessions we analyzed high-resolution RFs, the width and height, $w \times h$, of the ROI was $70 \times 70$ pixels, where each pixel subtends 1.6 arcminutes.

## Neural eye tracker calibration
The network used for calibrating the eye consisted of three parts: a *core* neural network that forms a nonlinear basis computed on the stimulus movies, a *readout* for each neuron that maps from the nonlinear features to spike rate, and a *shifter network* that shifts the stimulus using the measured gaze position. The architecture here was based on networks that have previously been successful for modeling V1 responses[35,48]. The neural network machinery in this application enabled us to optimize the weights in the shifter network and establish correction grids that shift the eye tracker's output into a more accurate estimate of gaze position.

The CNN core consists of a four-layer neural network. Each layer has a 2D convolutional stage, followed by a rectified linear (ReLU) function and Batch normalization. Time was embedded in the stimulus using the channel dimension. The number of channels per layer was 20 and the convolutional kernel sizes were 11, 9, 7, and 7 for the 4 layers respectively. All convolutions were windowed with a hamming window to avoid aliasing in the CNN core. We found that the shifter fitting was robust to the choice of core architecture but using smaller models that are more standard in neuroscience required optimizing hyperparameters for the subunit filters, whereas standard CNN architectures performed well without regularization.

The readout stage maps activations of the core to the spike rate of each neuron through an instantaneous affine transformation. To reduce the number of parameters and make the readout interpretable with respect to space, we used a factorized readout where each neuron, $i$, has a vector of feature weights that correspond to the 20 output channels from the final layer of the core, and a set of spatial weights.

The shifter network consists of a 2-layer network with 20 hidden units with SoftPlus activation functions in the first layer and 2 linear units in the second. It takes the measured gaze position on each frame that was used for the stimulus reconstruction and outputs a horizontal and vertical shift. The output of the shifter network was constrained to be 0 for the gaze position 0,0. The output of the shifter is used to sample from the stimulus using affine_grid and grid_sample functions in Pytorch. All parameters were learned simultaneously by minimizing

the Poisson loss using stochastic gradient descent with the Adam optimizer[49]. A Pytorch implementation with examples is available at (https://github.com/VisNeuroLab/yates-beyond-fixation).

The shifter calibration matrices were constructed by passing in a grid of potential gaze positions from −5 to 5 d.v.a centered on the center of the screen. These 2D correction grids were then used to correct the stimulus on each frame for gaze positions within that region. Specifically, the measured gaze position on each frame corresponds to a location in the correction grid. That location corresponds to an amount of shift. We used bilinear interpolation to map from gaze position to shift amount in the grid. The use of correction grids maps the two-layer shifter network into an interpretable format and these can be used across multiple stimulus sets to correct both the measured gaze position and the gaze-contingent stimulus reconstruction.

### High-resolution receptive fields

Receptive fields were recovered for the high-resolution stimuli using spike-triggered average (STA) on the pixels of reconstructed stimulus. The stimuli used for these RFs was either a sparse noise or Gabor noise stimulus. The sparse noise consisted of .1 d.v.a diameter black or white dots positioned randomly on each frame with a density of 1.5 dots/deg$^2$ on each frame. The Gabor noise consisted of multiscale Gabor patches with carrier frequencies ranging from 1 to 8.5 cycles per degree and widths (standard deviation of Gaussian) ranging from 0.05 to 0.132 d.v.a with a density of 2 Gabors/deg$^2$. The STA was computed 12 lags at 120 frames per second:

$$STA(\tau) = \frac{1}{N} \sum_t S(t - \tau) \qquad (5)$$

where $N$ is the number of spikes, $t$ is the frame index for each spike, $S$ is the stimulus frame, and $\tau$ is the lag. STAs were $z$-scored for visualization with the same normalizing constants before and after calibration.

### Reporting summary

Further information on research design is available in the Nature Portfolio Reporting Summary linked to this article.

## Data availability

Full and preprocessed datasets are available from the corresponding author (yates@berkeley.edu) upon reasonable request. An example dataset is available at https://doi.org/10.6084/m9.figshare.22580566 Source data are provided with this paper.

## Code availability

Code used to generate the visual stimuli and analyze the data are all available on GitHub in the following repositories: https://github.com/jcbyts/MarmoV5 and https://github.com/VisNeuroLab/yates-beyond-fixation.

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

## Acknowledgements

National Institutes of Health grants R00EY032179 (J.Y.), R01EY018363 (M.R.), R21EY025403 (D.B.) and R01EY030998 (J.M.). J.Y. was an Open Philanthropy fellow of the Life Sciences Research Foundation. National Science Foundation grants IIS 2113197 (D.B.), GRF DGE1745016 and DGE2140739 (G.S.). We thank Amelia Wattenberger for help with Figs. 1 and 4.

## Author contributions

J.Y. and J.M. designed the experiments. J.Y., R.W., and M.R. developed eye tracking methods. J.Y., S.C., G.S., and J.M. performed neurophysiology experiments. J.Y., D.B., and J.M. developed analyses and analyzed data. J.Y. and J.M. wrote the manuscript. All authors participated in revising the text.

## Competing interests

The authors declare no competing interests.
