## [Peer Review File · Nature Communications]

Detailed characterization of neural selectivity in free viewing primatesREVIEWER COMMENTS

Reviewer #1 (Remarks to the Author):

This is a creative and exceptionally valuable piece of work that tackles and goes a long way toward solving one of the long-standing and vexing problems of visual neuroscience — how to know how visual responses depend on neuronal tuning properties and eye movements. Framed another way, what are visual tuning properties inferred during eye movements? This is elegant, powerful work and I have only minor comments and questions.

1. A central claim of the paper is that measured RF properties are similar under fixation and during free viewing. This conclusion is well supported by comparison with existing literature, but it would be made even more solid if one could compare RF properties from the same cell under the two conditions. I understand that it is difficult to train marmosets to fixate, but is there no data that can be found to address this question? It would nail down a last unknown, the relationship between recovered receptive fields in a free foraging task and those observed during fixation. Differences between the two might be interesting, including if there were none. Tantalizingly now, we don't truly know.

2. I'm slightly concerned that the methods used might suffer from some subtle dynamic optical distortions. I have in mind the old studies of Deubel and Bridgeman ([https://doi.org/10.1016/0042-6989\(94\)00146-D](https://doi.org/10.1016/0042-6989(94)00146-D)) comparing eye traces recorded with coils vs a dual purkinje eye tracker. They found a systematic tendency for the fourth purkinje image to oscillate after a saccade. This is because the lens is elastic, it initially lags a saccade, then overshoots and takes some time for fibres of the zonule of Zinn to settle back to equilibrium. Maybe this isn't an issue but it takes about 30 ms (so about 3 frames) for things to stabilize, but the size of the retinal image displacement could be as large as 0.5 degrees. That doesn't only make things difficult in the fovea, but also in the parafovea to recover receptive field structure. Moreover, it would be dependent on the accommodation state of the subject (and their age obviously). This of course is not a defense of scleral eye coils, it is a question about whether the active paradigm might introduce constant artifacts. It would be helpful to see some consideration of this issue.

3. One specific problem that might emerge from issue 2 is that this effect might transiently scramble the estimated position of a target over a RF. If the target is a grating, then this amounts to phase jitter which might lead to an underestimate of the fraction of simple cells in the V1 sample — the proportion of simple cells would be underestimated. What is the simple vs complex cell breakdown in these V1 recordings, and how does it relate to measurements under fixation or paralysis? This is a useful validation of the method.

Reviewer #2 (Remarks to the Author):

This paper describes V1 receptive fields and MT direction tuning in free viewing marmosets. As the authors argue, most visual research is conducted under the constraint of fixation, even though animals normally move their eyes. Although this paper mainly confirms previous findings, it contains important methodological and proof-of-concept. It is hoped that this paper will contribute to new breakthroughs by combining this method with more natural scenes and new analyses. My comments are provided below.

Main comments

1. Figure 1 shows a model of a V1 simple cell. However, the analysis in Figure 2 show complex cell receptive fields. Why is this? It is necessary to describe whether it is possible to reconstitute the receptive field of simple cells.
2. More description of the population is needed to assess how effective this method is. For example, the V1 receptive field is described as recovered in 410 of 739 units. This means that nearly half were not recovered. Is this a failure of the method? Or is this consistent with previous studies under fixation? Needless to say, a population description is necessary for the MT and high-resolution foveal receptive field analyses as well.
3. The CNN analysis is difficult to understand and evaluate. I wish there was a way to prove that this is real. For example, if you shuffle the timing of spikes (a permutation test), does the CNN not give a consistent spatial configuration of the receptive field? This is to confirm that the CNN does not create something out of nothing.
4. I would like a more detailed description of the second surgical procedure. Did you use only Ketamine and Dexmedetomidine for the craniotomy?

Minor comments

1. How many neurons and how many sessions were obtained in each monkey? Please describe for all experiments (V1 receptive field, V1 grating, MT, and high-resolution foveal receptive fields).

2. Figure 3 only shows 3 examples. Was direction tuning reconstructed for all 21 neurons?
3. How often were targets presented in the foraging task?
4. What was the luminance of the dots and screen?
5. The equation seems to be missing on line 494.
6. Supplementary Figure 2. This is an important figure and could go to the main figure.
7. Supplementary Figure 2. Why is $n=426$ for a and $n=471$ for b? In the description of the results, it says that 410 neurons recovered receptive fields and 437 neurons recovered selective responses to full-field gratings.
8. Supplementary Figure 2. There seems to be a bias in orientation preferences. It would be helpful if you could cite previous papers describing this “oblique effect” (e.g., Li, Peterson & Freeman, 2003, J Neurophysiol) to evaluate the proposed method.
9. Supplementary Figure 2. The authors claim that the results are comparable to previous papers by inspecting polynomial fit (a) or running mean (c). Is the variability in data points also comparable to previous reports?

Reviewer #1 (Remarks to the Author):

This is a creative and exceptionally valuable piece of work that tackles and goes a long way toward solving one of the long-standing and vexing problems of visual neuroscience — how to know how visual responses depend on neuronal tuning properties and eye movements. Framed another way, what are visual tuning properties inferred during eye movements? This is elegant, powerful work and I have only minor comments and questions.

Thanks for the nice comments. We are also excited about this work and the directions it opens.

1. A central claim of the paper is that measured RF properties are similar under fixation and during free viewing. This conclusion is well supported by comparison with existing literature, but it would be made even more solid if one could compare RF properties from the same cell under the two conditions. I understand that it is difficult to train marmosets to fixate, but is there no data that can be found to address this question? It would nail down a last unknown, the relationship between recovered receptive fields in a free foraging task and those observed during fixation. Differences between the two might be interesting, including if there were none. Tantalizingly now, we don't truly know.

Thank you for the thoughtful question. One of the central scientific motivations for this methodological shift from fixation to free-viewing is that the input generated by eye movements will modulate neural responses – in fact, we believe eye movements are a central part of the neural code for vision and have further studies planned to address this question more broadly. However, implicit in our writing of this manuscript is the idea that receptive fields will be the “same” during fixation periods of an active foraging task, as compared to traditional approaches with prolonged fixation periods. Exactly how similar does an RF need to be to call it the same? Because it is known that RF properties change with stimulus statistics (1) and eye movements modify the stimulus statistics (2) we would expect some differences between free-viewing and fixation due to changes in the retinal input. Therefore, there are probably many RFs that can be measured for a given neuron, depending on the stimulus used, and we'd argue that free-viewing can recover as good a measurement of “the RF” as during fixation. Nonetheless, we agree that it is important to have a direct comparison of fixation and free-viewing in the same neurons.

Therefore, to address your question, we ran additional experiments where the same foveal neurons were held during fixation and free viewing. Interestingly, the center of gaze drifts during fixation by much more than the size of foveal receptive fields, which would be an obstacle even for recordings with macaques that can perform prolonged fixation tasks. We found that it was impossible to measure a foveal RF during fixation in the marmoset using conventional approaches, which is not different from previous attempts with macaques (3–5). However, if we do correct for fixational eye movements in our analyses during the task in which a marmoset is fixating, then we are able to recover receptive fields and compare them to those receptive fields we find during free-viewing tasks. Under those conditions we do find that the simplest measure of the receptive field, the linear component represented by the spike triggered average (STA), is well correlated between the two methods. This is also an exciting point that highlights the value

of our method: while we cannot measure foveal RFs in a fixation task using a traditional approach that ignores small fixational eye movements, once we correct offline for eye movements then it becomes possible, and the resulting estimate is actually no better than if we had done so during free-viewing task, a task that requires no or minimal effort for the observer. We have included an additional supplemental figure about this point.

Supplemental Figure 5, reproduced below, shows the receptive fields of example neurons recovered during fixation, fixation with gaze correction, and free viewing. We have also included this point in the main text which reads as:

Line 311-315 of main text and supplemental figure 5 are reproduced here:

To compare the RFs recovered during free-viewing with fixation more directly, we recorded the same units during fixation and free-viewing. Foveal RFs were too small to be recovered during standard fixation without correcting for measured eye movements. Further, we found a close correspondence between the RF recovered during fixation with eye correction and free viewing (Supplemental Figure 5).

Supplemental Figure 5 | Comparison of the same units during fixation and free viewing.

a. Example cells spatial RFs (STAs) measured during free-viewing (FV), fixation with correction for measured eye movements (FX) and fixation with no correction (FX (raw)). Contours indicate the region that will be correlated to compare across condition. **b.** Scatter plot of RF similarity within and across conditions. The x-axis is the within-condition correlation plotted for subsets of

the fixation data. The y-axis is the RF correlation between fixation and free viewing. The medians are not significantly different ($p = 0.31$, statistic=820.0, Mann-Whitney U test) **c.** Same as b, except the y axis is the correlation between fixation with correction and fixation without correction. The medians are significantly different ($p < 0.001$, statistic=820) **d.** same as b,c except the y axis is the within correlation for subsets of free-viewing data. The medians are significantly different ($p < 0.001$, statistic=402.0, Mann-Whitney U test). The fact that the within correlation is higher for free-viewing suggests we get a more consistent RF in that condition than using fixation.

2. I'm slightly concerned that the methods used might suffer from some subtle dynamic optical distortions. I have in mind the old studies of Deubel and Bridgeman ([https://doi.org/10.1016/0042-6989\(94\)00146-D](https://doi.org/10.1016/0042-6989(94)00146-D)) comparing eye traces recorded with coils vs a dual purkinje eye tracker. They found a systematic tendency for the fourth purkinje image to oscillate after a saccade. This is because the lens is elastic, it initially lags a saccade, then overshoots and takes some time for fibres of the zonule of Zinn to settle back to equilibrium. Maybe this isn't an issue but it takes about 30 ms (so about 3 frames) for things to stabilize, but the size of the retinal image displacement could be as large as 0.5 degrees. That doesn't only make things difficult in the fovea, but also in the parafovea to recover receptive field structure. Moreover, it would be dependent on the accommodation state of the subject (and their age obviously). This of course is not a defense of scleral eye coils, it is a question about whether the active paradigm might introduce constant artifacts. It would be helpful to see some consideration of this issue.

We do indeed see evidence of post-saccadic lens wobble in our digital Dual Purkinje eye traces and were concerned about this as well. In fact, we did include some control to mitigate this effect in our analyses, although it probably was not emphasized enough or the reasoning behind it. In fact, the high-resolution analyses in the manuscript begin 50ms after saccade offset. We now cite the Deubel and Bridgeman reference to highlight the issue with the dDPI eye tracker, and then also emphasize in the main text that we have omitted those times adjacent to the saccades in our analysis of receptive fields (line 282).

3. One specific problem that might emerge from issue 2 is that this effect might transiently scramble the estimated position of a target over a RF. If the target is a grating, then this amounts to phase jitter which might lead to an underestimate of the fraction of simple cells in the V1 sample — the proportion of simple cells would be underestimated. What is the simple vs complex cell breakdown in these V1 recordings, and how does it relate to measurements under fixation or paralysis? This is a useful validation of the method.

In the present manuscript we had aimed to demonstrate the potential of the free-viewing method. Therefore, we focused on the simplest receptive field analysis, recovering the spike-triggered average, to emphasize that sophisticated models are not necessary to recover meaningful properties once offline correction for eye movements have been implemented.

Nonetheless, our method does afford the opportunity to recover more complex receptive fields, and in line with the reviewer's question, we can address the proportion of simple versus complex cells in this method. We do not find the proportion to be at odds with previous literature, supporting that any potential artifacts from lens wobble have been omitted from our analyses by only analyzing periods of stable fixation well after saccade offset.

To estimate complex receptive fields we relied on a model comparison between an energy model fit to the data versus a simple linear nonlinear Poisson model, both including a static non-linearity for the transform to spike rates. This type of model comparison has been shown to track the simple-complex distinction well (6). We have included these results as supplemental figure 4 (reproduced below). These results are roughly consistent with the proportions reported from previous studies from anesthetized animals (7). We have included a supplemental figure (reproduced below) to report on these effects and mention the results briefly in the main text (line 296-302) to consider if variation in position from free-viewing might increase phase scramble, for which we do not find support.

The text from lines 296-302 and supplemental figure 4 are reproduced here:

Because errors in measured gaze position may manifest as a scrambling of the phase of the input, which would result in a bias towards finding complex cells, we investigated the proportion of simple and complex cells in our sample by fitting simple and complex cell models to each cell and comparing the cross-validated log-likelihoods (Supplemental Figure 4). Using this method, we found 81% of our units were complex, consistent with previous reports from anesthetized marmoset measured outside the fovea (37).

Supplemental Figure 4 | Simple and complex cells. **a.** Schematic of a simple cell and complex cell model. Following Vintch et al., 2015, we fit a linear nonlinear Poisson model (LNP) and an Energy Model with quadratic subunits followed by a static nonlinearity and Poisson process. Both models were fit directly to spike counts by minimizing the Poisson loss with L-BFGS. **b.** Log-likelihood (bits/spike) on withheld data for the Energy model and LNP model. Cells to the right of the unity line were classified as simple cells. This classification has been shown to track classic measures of simple and complex cells well (Vintch et al., 2015). Example cells from panel c are highlighted. **c.** Subunits visualized for three example cells shown in panel b. (top row) the weights from the LNP model visualized as the spatial RF at the peak lag and the temporal dynamics of the weights visualized by plotting the spatial locations with the maximum (blue) and minimum (red) values at all time lags. (bottom rows) subunit weights visualized in the

same way as the linear weights above. Each column shows the subunits for the same example cell shown in the top row. Example cell 1 is classified as a simple cell. Although there is spatial structure in its nonlinear subunits, the temporal structure is noisy and the model overfits the training data and cross-validates worse than the LNP model.

Reviewer #2 (Remarks to the Author):

This paper describes V1 receptive fields and MT direction tuning in free viewing marmosets. As the authors argue, most visual research is conducted under the constraint of fixation, even though animals normally move their eyes. Although this paper mainly confirms previous findings, it contains important methodological and proof-of-concept. It is hoped that this paper will contribute to new breakthroughs by combining this method with more natural scenes and new analyses. My comments are provided below.

Main comments

1. Figure 1 shows a model of a V1 simple cell. However, the analysis in Figure 2 shows complex cell receptive fields. Why is this? It is necessary to describe whether it is possible to reconstitute the receptive field of simple cells.

We apologize for any confusion that the model used for illustration in Figure 1c may have caused and have updated the manuscript to clarify our goals, which are two-fold: first, that in free-viewing much of the complex RF properties can be recovered even without high precision and accuracy in eye tracking (Figures 2 and 3), and second, with the addition of advanced eye tracking, even the most precise RF structures epitomized by simple cells linear RF kernels can be recovered (Figure 4). Figure 4 and the newly-added Supplemental Figure 4 show that this method reconstructs linear RFs, including those of simple cells. We have revised the main text to highlight this point in the transition from introduction to results by moving the second section title to begin just before Figure 2 and clarifying our goal in this section as recovering coarse retinotopy and feature tuning with standard eye tracking. This revised transition occurs after Figure 1c and reads on lines 118-121 as:

“Retinotopy and selectivity in V1 during free-viewing paradigms.

While the accuracy and precision of eye tracking impose limitations on the scale of receptive fields that can be studied during free viewing, a great deal can still be accomplished even with a standard eye tracker. In this section, we show that full-field sparse noise stimuli and commercially available eye tracking (Eyelink 1000) can be used to recover the size, location and tuning of receptive fields (RFs) ...”

This opens our discussion of Figure 2 where we demonstrate that one can recover retinotopic location using coarse retinotopic flashed stimuli (large sparse noise dots) as well as tuning for orientation and spatial frequency with full-field flashed gratings (i.e., following sub-space reverse correlation methods (8)). The value of showing RF recovery in these feature spaces is possible

in that it does not require sophisticated eye tracking, and thus would be readily available to a wide audience and enable them to rapidly get estimates of the retinotopy and neuronal tuning from free-viewing tasks. Further, this method generalizes beyond V1 to, for example, recover MT receptive field structure and tuning for motion (Figure 3). But there are some limitations when eye tracking is coarse. To recover precise receptive field structure at the level of simple cells, as illustrated in Figure 1 and then later from our data in Figure 4 and Supplemental Figures 4 and 5, it does absolutely require both high precision and accuracy in eye tracking.

2. More description of the population is needed to assess how effective this method is. For example, the V1 receptive field is described as recovered in 410 of 739 units. This means that nearly half were not recovered. Is this a failure of the method? Or is this consistent with previous studies under fixation? Needless to say, a population description is necessary for the MT and high-resolution foveal receptive field analyses as well.

We have expanded discussion of the proportion of cells recovered throughout the manuscript. The proportion we recover is consistent with some previous studies that have used fixation with comparable recording probes in V1 (9), but it is also important to recognize that the number of cells recovered depends highly both on the method of recording (single unit tungsten versus silicon arrays and how spike sorting is performed) and also on the type of visual stimulus employed (low dimensional space such as coarse dots or orientation/spatial frequency, as in Figure 2, as opposed to natural image stimuli or Gabors varying in size, spatial frequency and orientation, which span a larger space and can activate more cells that would otherwise be silent). Last, but not least, recovery depends on the firing rate of individual neurons and on the amount of recording time that the neuron can be isolated.

The purpose of our present study was not to directly compare proportions of cells recovered with previous literature or to claim that our method recovers more than previous methods, as those outcomes will depend on a variety of complex factors. Rather our main goal is to demonstrate the utility of free-viewing for learning about visual neurons receptive field properties without the constraint of a fixation task. We have added statistics for the recovery proportions of simple and complex cell receptive fields regarding analyses from Figure 4 and added a new supplemental figure to present those results (supplemental figure 4, see response to reviewer 1 above). We have further included 3 supplemental tables that detail each session that contributes to each analysis and the resulting number of recovered receptive fields using our criterion. You'll note that the number of total units has decreased in supplemental figure 2. This is because we had previously included sessions in the coarse-mapping section (Figure 2) where the sparse noise stimuli and grating stimuli were presented for very short periods of time (< 1 minute). As those mapping sessions were not intended to reconstruct individual neuron RFs, and the recording time was insufficient to do so, it was an error to include them in our original submission. We have updated our inclusion for minimal recording time accordingly in methods. We hope this clarifies which cells are included in the analyses that support each figure.

3. The CNN analysis is difficult to understand and evaluate. I wish there was a way to prove that this is real. For example, if you shuffle the timing of spikes (a permutation test), does the CNN not give a consistent spatial configuration of the receptive field? This is to confirm that the CNN does not create something out of nothing.

We recognize the weakness of results relying on a CNN analysis. In fact, this is part of the reason why we had chosen to focus on simple receptive fields using the very most basic of analyses, the spike triggered average, in Figure 4. It is important to understand that these simple receptive fields do NOT emerge from the hidden units of the CNN. We have revised the main text to further clarify this point on lines 257-260 and 364-367. The CNN described in Figure 4a has only one role and that is recalibrate the eye tracker. It enables us to correct for inaccuracies in the human experimenter's initial calibration (such as horizontal and vertical gain) but also more subtle changes such as inaccuracies in fixation during calibration. The validity of the CNN towards improving the accuracy of our eye tracking is then revealed when we compare the improvement in recovered STAs (for comparison with and without the improved eye tracking, see Figure 4c). Furthermore, the receptive fields shown (Figure 4d) are normalized (z-scored) based on the variance outside the RF, so they reflect the confidence in the estimates and demonstrate they are not arbitrary noise. A further validation is that they remain consistent across statistically independent lags of video, which we have demonstrated by showing the full spatiotemporal RF rather than just a single optimal delay. We have revised the main text to emphasize this point better and explain the statistical significance of the RFs shown in Figure 4.

4. I would like a more detailed description of the second surgical procedure. Did you use only Ketamine and Dexmedetomidine for the craniotomy?

Yes. That's correct. This drug cocktail provides a deep plane of anesthesia in marmosets for approximately 45-90 minutes, which is sufficient to perform a craniotomy in our hands.

Minor comments

1. How many neurons and how many sessions were obtained in each monkey? Please describe for all experiments (V1 receptive field, V1 grating, MT, and high-resolution foveal receptive fields).

Thanks for the suggestion. We have added three tables in the supplementary materials to summarize this information.

2. Figure 3 only shows 3 examples. Was direction tuning reconstructed for all 21 neurons?

Yes. We had intended this only as a proof of principle for the method, but have now included additional statistics from a more complete population of 241 units recorded across 10 sessions. This point has been made clear in the revised manuscript on lines 199-202:

Following the selection criterion above, we labeled neurons as having selective RFs if the linear RF cross-validated better than the mean firing rate. This yielded 241/466 (51.72%) selective units (Supplemental Table 2). The selective MT neurons in our sample were well fit by von Mises tuning curves (mean r-squared = 0.62 ± 0.02, n=241 units).

3. How often were targets presented in the foraging task?

A single face or Gabor target was presented at all times. Once the target was acquired for liquid reward it was immediately replotted at a new location allowing the task to continue. This statement has been added to the methods (line: 496-498).

4. What was the luminance of the dots and screen?

The luminance of the white dots was 854 cd/m², the black dots were 0.49 cd/m² and the gray background was 416 cd/m². We have updated this in methods describing the projector and dots (line 483 and line 509).

5. The equation seems to be missing on line 494.

We apologize for our error and have included the equation (now on line 545-547).

6. Supplementary Figure 2. This is an important figure and could go to the main figure.

We thank the reviewer for the suggestion and would be willing to include Supplementary Figure 2 as part of the panels for Figure 2 in the main text if requested. We were concerned that it lengthens the manuscript for the general audience and interpreting it more broadly involves nuance for several technical points that we will outline here: First, supplemental Figure 2 combines data across several monkeys and does not have a thorough exploration of eccentricity in each monkey. Second, we believe these analyses should be conducted at the full pixel resolution as in figure 4 and have experiments planned to complete this for future studies. Third, as you will see in the response above, many of the units that previously contributed to supplemental figure 2 have been excluded now because they were collected using very brief experimental durations. These brief durations were used simply to map the retinotopy of the recording site and do not provide sufficient statistical power to map every individual unit collected.

7. Supplementary Figure 2. Why is n=426 for a and n=471 for b? In the description of the results, it says that 410 neurons recovered receptive fields and 437 neurons recovered selective responses to full-field gratings.

The distinction between panel a and b is that RF size must be estimated from the coarse spatial mapping with sparse noise, for which not all neurons produced a significant receptive field in the time available for that mapping procedure. The data in panel b-c derives from mapping using flashed gratings, which produced more robust responses and enabled more neurons to be

recovered. Another reason has to do with the total number of spikes produced by each neuron under each condition. Without sufficient spikes, we do not have the ability to recover RFs. We have revised description of the results to include those details and included a supplemental table for both conditions that explain how these numbers come from our spike threshold and RF analyses.

8. Supplementary Figure 2. There seems to be a bias in orientation preferences. It would be helpful if you could cite previous papers describing this “oblique effect” (e.g., Li, Peterson & Freeman, 2003, J Neurophysiol) to evaluate the proposed method.

Thank you for the suggestion. We have added this reference (line 177).

9. Supplementary Figure 2. The authors claim that the results are comparable to previous papers by inspecting polynomial fit (a) or running mean (c). Is the variability in data points also comparable to previous reports?

Thank you for raising this point. The variability in RF size is larger than that reported in the original Rosa study. Some of this resulted from an error on our part by including some sessions that have very brief (<1 minute) recording durations that were not intended to map individual neurons RFs (see above). We have removed those sessions in the updated Supplemental Figure. We reported that the results are comparable by looking at the confidence intervals on the fitted parameters of the polynomial from Rosa et al., 1997. Their parameters and our fits are now reported in the Supplemental Figure.

We suspect the main differences from the Rosa study and ours has to do with how neurons were sampled. First, the Rosa study recorded from anesthetized marmosets where the numbers and types of neurons activated may have differed from the awake prep as in our study. Second, they recorded using tungsten electrodes in which well-isolated single units were manually identified, likely introducing a sampling bias towards larger spike waveform cells with higher firing rates. By contrast, our recordings are taken from linear arrays that include any neuron with identified spikes, thus sampling a larger population including neurons with low firing rates. Third, They also employed hand mapping procedures rather than quantification using reverse correlation, which may have introduced some experimenter bias (e.g., to impose some prior knowledge on noisy RFs based on the estimates from surrounding neurons). As a result, we expect higher variability in our RF sizes both because we sample from larger populations, but also because we likely include neurons with lower firing rates that will give more variable estimates of RF structure due to having fewer spikes. These details further cement our feelings that Supplemental Figure 2 should remain in the supplement, but we are happy to reconsider and include these points in discussion if you feel the manuscript would benefit from them.

Bibliography

1. Almasi A, Sun SH, Yunzab M, Jung YJ, Meffin H, Ibbotson MR. How stimulus statistics affect the receptive fields of cells in primary visual cortex. *J Neurosci*. 2022 Jun 29;42(26):5198–211.
2. Rucci M, Victor JD. The unsteady eye: an information-processing stage, not a bug. *Trends Neurosci*. 2015 Apr;38(4):195–206.
3. McFarland JM, Bondy AG, Cumming BG, Butts DA. High-resolution eye tracking using V1 neuron activity. *Nat Commun*. 2014 Sep 8;5:4605.
4. McFarland JM, Cumming BG, Butts DA. Variability and correlations in primary visual cortical neurons driven by fixational eye movements. *J Neurosci*. 2016 Jun 8;36(23):6225–41.
5. Snodderly DM. A physiological perspective on fixational eye movements. *Vision Res*. 2016 Jan;118:31–47.
6. Vintch B, Movshon JA, Simoncelli EP. A convolutional subunit model for neuronal responses in macaque V1. *J Neurosci*. 2015 Nov 4;35(44):14829–41.
7. Yu H-H, Rosa MGP. Uniformity and diversity of response properties of neurons in the primary visual cortex: selectivity for orientation, direction of motion, and stimulus size from center to far periphery. *Vis Neurosci*. 2014 Jan;31(1):85–98.
8. Ringach DL, Shapley RM, Hawken MJ. Orientation selectivity in macaque V1: diversity and laminar dependence. *J Neurosci*. 2002 Jul 1;22(13):5639–51.
9. De A, Horwitz GD. Spatial receptive field structure of double-opponent cells in macaque V1. *J Neurophysiol*. 2021 Mar 1;125(3):843–57.

REVIEWERS' COMMENTS

Reviewer #1 (Remarks to the Author):

The authors have responded well to my comments and suggestions. I have nothing further.

Reviewer #2 (Remarks to the Author):

The authors have adequately addressed all of my comments. I commend the authors for their diligent efforts. I have no further comments.

REVIEWERS' COMMENTS

Reviewer #1 (Remarks to the Author):

The authors have responded well to my comments and suggestions. I have nothing further.

Reviewer #2 (Remarks to the Author):

The authors have adequately addressed all of my comments. I commend the authors for their diligent efforts. I have no further comments.

We thank the reviewers for their feedback.